# Counterfactual Explanations Can Be Manipulated

**Dylan Slack**
UC Irvine
dslack@uci.edu

**Sophie Hilgard**
Harvard University
ash798@g.harvard.edu

**Himabindu Lakkaraju**
Harvard University
hlakkaraju@hbs.edu

**Sameer Singh**
UC Irvine
sameer@uci.edu

## Abstract

Counterfactual explanations are emerging as an attractive option for providing recourse to individuals adversely impacted by algorithmic decisions. As they are deployed in critical applications (e.g. law enforcement, financial lending), it becomes important to ensure that we clearly understand the vulnerabilties of these methods and find ways to address them. However, there is little understanding of the vulnerabilities and shortcomings of counterfactual explanations. In this work, we introduce the first framework that describes the vulnerabilities of counterfactual explanations and shows how they can be manipulated. More specifically, we show counterfactual explanations may converge to drastically different counterfactuals under a small perturbation indicating they are not robust. Leveraging this insight, we introduce a novel objective to train seemingly fair models where counterfactual explanations find much lower cost recourse under a slight perturbation. We describe how these models can unfairly provide low-cost recourse for specific subgroups in the data while appearing fair to auditors. We perform experiments on loan and violent crime prediction data sets where certain subgroups achieve up to 20x lower cost recourse under the perturbation. These results raise concerns regarding the dependability of current counterfactual explanation techniques, which we hope will inspire investigations in robust counterfactual explanations.[1]

## 1 Introduction

Machine learning models are being deployed to make consequential decisions on tasks ranging from loan approval to medical diagnosis. As a result, there are a growing number of methods that explain the decisions of these models to affected individuals and provide means for *recourse* [1]. For example, recourse offers a person denied a loan by a credit risk model a reason for *why* the model made the prediction and *what can be done* to change the decision. Beyond providing guidance to stakeholders in model decisions, algorithmic recourse is also used to detect discrimination in machine learning models [2–4]. For instance, we expect there to be minimal *disparity* in the *cost* of achieving recourse between both men and women who are denied loans. One commonly used method to generate recourse is that of *counterfactual explanations* [5]. Counterfactual explanations offer recourse by attempting to find the minimal change an individual must make to receive a positive outcome [6–9].

Although counterfactual explanations are used by stakeholders in consequential decision-making settings, there is little work on systematically understanding and characterizing their limitations. Few recent studies explore how counterfactual explanations may become valid when the underlying model is updated. For instance, a model provider might decide to update a model, rendering previously generated counterfactual explanations invalid [10, 11]. Others point out that counterfactual explanations, by ignoring the causal relationships between features, sometimes recommend changes that are not actionable [12]. Though these works shed light on certain shortcomings of counterfactual explanations, they do not consider whether current formulations provide stable and reliable results, whether they can be manipulated, and if fairness assessments based on counterfactuals can be trusted.

---

[1]Project Page: https://dylanslacks.website/cfe/

35th Conference on Neural Information Processing Systems (NeurIPS 2021).

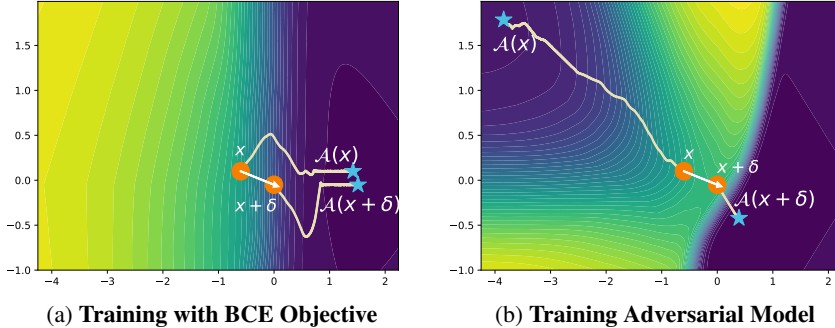

| (a) **Training with BCE Objective** | (b) **Training Adversarial Model** |

Figure 1: **Model trained with BCE objective and adversarial model on a toy data set** using Wachter et al.'s Algorithm [6]. The surface shown is the loss in Wachter et al.'s Algorithm with respect to $x$, the line is the path of the counterfactual search, and we show results for a single point, $x$. For the model without the manipulation (subfigure 1a), the counterfactuals for $x$ and $x + \delta$ converge to the same minima and are similiar cost recourse. For the adversarial model (subfigure 1b), the recourse found for $x$ has *higher cost* than $x + \delta$ because the local minimum initialized at $x$ is *farther* than the minimum starting at $x + \delta$, demonstrating the problematic behavior of counterfactual explanations.

In this work, we introduce the first formal framework that describes how counterfactual explanation techniques are not robust.[2] More specifically, we demonstrate how the family of counterfactual explanations that rely on hill-climbing (which includes commonly used methods like Wachter et al.'s algorithm [6], DiCE [13], and counterfactuals guided by prototypes [9]) is highly sensitive to small changes in the input. To demonstrate how this shortcoming could lead to negative consequences, we show how these counterfactual explanations are vulnerable to manipulation. Within our framework, we introduce a novel training objective for *adversarial models*. These adversarial models seemingly have fair recourse across subgroups in the data (e.g., men and women) but have much lower cost recourse for the data under a slight perturbation, allowing a bad-actor to provide low-cost recourse for specific subgroups simply by adding the perturbation. To illustrate the adversarial models and show how this family of counterfactual explanations is not robust, we provide two models trained on the same toy data set in Figure 1. In the model trained with the standard BCE objective (left side of Fig 1), the counterfactuals found by Wachter et al.'s algorithm [6] for instance $x$ and perturbed instance $x + \delta$ converge to same minima (denoted $\mathcal{A}(x)$ and $\mathcal{A}(x + \delta)$). However, for the adversarial model (right side of Fig 1), the counterfactual found for the perturbed instance $x + \delta$ is *closer* to the original instance $x$. This result indicates that the counterfactual found for the perturbed instance $x + \delta$ is *easier to achieve* than the counterfactual for $x$ found by Wachter et al.'s algorithm! Intuitively, counterfactual explanations that hill-climb the gradient are susceptible to this issue because optimizing for the counterfactual at $x$ versus $x + \delta$ can converge to different local minima.

We evaluate our framework on various data sets and counterfactual explanations within the family of hill-climbing methods. For Wachter et al.'s algorithm [6], a sparse variant of Wachter et al.'s, DiCE [13], and counterfactuals guided by prototypes [9], we train models on data sets related to loan prediction and violent crime prediction with fair recourse across subgroups that return 2-20× lower cost recourse for specific subgroups with the perturbation $\delta$, without any accuracy loss. Though these results indicate counterfactual explanations are highly vulnerable to manipulation, we consider making counterfactual explanations that hill-climb the gradient more robust. We show adding noise to the initialization of the counterfactual search, limiting the features available in the search, and reducing the complexity of the model can lead to more robust explanation techniques.

## 2 Background

In this section, we introduce notation and provide background on counterfactual explanations.

---

[2]Note, that the usage of "counterfactual" does not have the same meaning as it does in the context of causal inference, and we adopt the term "counterfactual explanation" for consistency with prior literature.

**Notation** We use a dataset $\mathcal{D}$ containing $N$ data points, where each instance is a tuple of $\boldsymbol{x} \in \mathbb{R}^d$ and label $y \in \{0, 1\}$, i.e. $\mathcal{D} = \{(\boldsymbol{x}_n, y_n)\}_{n=1}^N$ (similarly for the test set). For convenience, we refer to the set of all data points $\boldsymbol{x}$ in dataset $\mathcal{D}$ as $\mathcal{X}$. We will use the notation $\boldsymbol{x}_i$ to denote indexing data points in $\mathcal{X}/\mathcal{D}$ and $\boldsymbol{x}^q$ to denote indexing attribute $q$ in $\boldsymbol{x}$. Further, we have a model that predicts the probability of the positive class using a datapoint $f : x \to [0, 1]$. Further, we assume the model is paramaterized by $\boldsymbol{\theta}$ but omit the dependence and write $f$ for convenience. Last, we assume the positive class is the desired outcome (e.g., receiving a loan) henceforth.

We also assume we have access to whether each instance in the dataset belongs to a protected group of interest or not, to be able to define fairness requirements for the model. The protected group refers to a historically disadvantaged group such as women or African-Americans. We use $\mathcal{D}_{\text{pr}}$ to indicate the protected subset of the dataset $\mathcal{D}$, and use $\mathcal{D}_{\text{np}}$ for the "not-protected" group. Further, we denote protected group with the positive (i.e. *more desired*) outcome as $\mathcal{D}_{\text{pr}}^{\text{pos}}$ and with negative (i.e. *less desired*) outcome as $\mathcal{D}_{\text{pr}}^{\text{neg}}$ (and similarly for the non-protected group).

**Counterfactual Explanations** Counterfactual explanations return a data point that is *close* to $\boldsymbol{x}$ but is predicted to be positive by the model $f$. We denote the counterfactual returned by a particular algorithm $\mathcal{A}$ for instance $\boldsymbol{x}$ as $\mathcal{A}(\boldsymbol{x})$ where the model predicts the positive class for the counterfactual, i.e., $f(\mathcal{A}(\boldsymbol{x})) > 0.5$. We take the difference between the original data point $\boldsymbol{x}$ and counterfactual $\mathcal{A}(\boldsymbol{x})$ as the set of changes an individual would have to make to receive the desired outcome. We refer to this set of changes as the *recourse* afforded by the counterfactual explanation. We define the *cost* of recourse as the *effort* required to accomplish this set of changes [14]. In this work, we define the cost of recourse as the distance between $\boldsymbol{x}$ and $\mathcal{A}(\boldsymbol{x})$. Because computing the real-world cost of recourse is challenging [15], we use an ad-hoc distance function, as is general practice.

**Counterfactual Objectives** In general, counterfactual explanation techniques optimize objectives of the form,

$$G(\boldsymbol{x}, \boldsymbol{x}_{\text{cf}}) = \lambda \cdot (f(\boldsymbol{x}_{\text{cf}}) - 1)^2 + d(\boldsymbol{x}, \boldsymbol{x}_{\text{cf}}) \tag{1}$$

in order to return the counterfactual $\mathcal{A}(\boldsymbol{x})$, where $\boldsymbol{x}_{\text{cf}}$ denotes *candidate* counterfactual at a particular point during optimization. The first term $\lambda \cdot (f(\boldsymbol{x}_{\text{cf}}) - 1)$ encourages the counterfactual to have the desired outcome probability by the model. The distance function $d(\boldsymbol{x}, \boldsymbol{x}_{\text{cf}})$ enforces that the counterfactual is close to the original instance and easier to "achieve" (lower cost recourse). $\lambda$ balances the two terms. Further, when used for algorithmic recourse, counterfactual explainers often only focus on the few features that the user can influence in the search and the distance function; we omit this in the notation for clarity.

**Distance Functions** The distance function $d(\boldsymbol{x}, \boldsymbol{x}_{\text{cf}})$ captures the *effort* needed to go from $\boldsymbol{x}$ to $\boldsymbol{x}_{\text{cf}}$ by an individual. As one such notion of distance, Wachter et al. [6] use the Manhattan ($\ell_1$) distance weighted by the inverse median absolute deviation (MAD).

$$d(\boldsymbol{x}, \boldsymbol{x}_{\text{cf}}) = \sum_{q \in [d]} \frac{|\boldsymbol{x}^q - \boldsymbol{x}_{\text{cf}}^q|}{\text{MAD}_q} \quad \text{MAD}_q = \text{median}_{i \in [N]} \left( |x_i^q - \text{median}_{j \in [N]}(x_j^q)| \right) \tag{2}$$

This distance function generates sparse solutions and closely represents the absolute change someone would need to make to each feature, while correcting for different ranges across the features. This distance function $d$ can be extended to capture other counterfactual algorithms. For instance, we can include elastic net regularization instead of $\ell_1$ for more efficient feature selection in high dimensions [16], add a term to capture the closeness of the counterfactual $\boldsymbol{x}_{\text{cf}}$ to the data manifold to encourage the counterfactuals to be in distribution, making them more realistic [9], or include diversity criterion on the counterfactuals [13]. We provide the objectives for these methods in Appendix B.1.

**Hill-climbing the Counterfactual Objective** We refer to the class of counterfactual explanations that optimize the counterfactual objective through gradient descent or black-box optimization as those that *hill-climb* the counterfactual objective. For example, Wachter et al.'s algorithm [6] or DiCE [13] fit this characterization because they optimize the objective in Equation 1 through gradient descent. Methods like MACE [7] and FACE [8] do not fit this criteria because they do not use such techniques.

**Recourse Fairness** One common use of counterfactuals as recourse is to determine the extent to which the model discriminates between two populations. For example, counterfactual explanations may return recourses that are easier to achieve for members of the not-protected group [1, 4] indicating

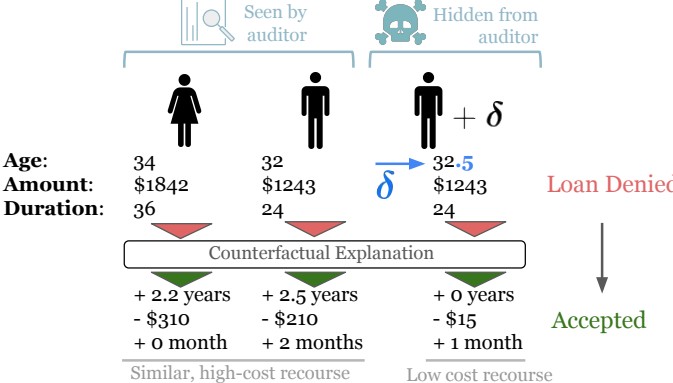

Figure 2: **Manipulated Model for Loan Risk.** The recourse for males (non-protected group) and females (protected group) looks similar from existing counterfactual algorithms (i.e. model seems fair). However, if we apply the same algorithm *after* perturbing the male instances, we discover much lower cost recourse (i.e. the model discriminates between sexes).

unfairness in the counterfactuals [3, 2]. Formally, we define the recourse fairness as the difference in the average distance of the recourse cost between the protected and not-protected groups and say a counterfactual algorithm $\mathcal{A}$ is *recourse fair* if this disparity is less than some threshold $\tau$.

**Definition 2.1** *A model $f$ is recourse fair for algorithm $\mathcal{A}$, distance function $d$, dataset $\mathcal{D}$, and scalar threshold $\tau$ if [2],*

$$\left| \mathbb{E}_{x \sim \mathcal{D}_{pr}^{neg}} \left[ d\left(\boldsymbol{x}, \mathcal{A}(\boldsymbol{x})\right) \right] - \mathbb{E}_{x \sim \mathcal{D}_{np}^{neg}} \left[ d\left(\boldsymbol{x}, \mathcal{A}(\boldsymbol{x})\right) \right] \right| \le \tau$$

# 3 Adversarial Models for Manipulating Counterfactual Explanations

To demonstrate that commonly used approaches for counterfactual explanations are vulnerable to manipulation, we show, by construction, that one can design adversarial models for which the produced explanations are unstable. In particular, we focus on the use of explanations for determining fair recourse, and demonstrate that models that produce seemingly fair recourses are in fact able to produce much more desirable recourses for non-protected instances if they are perturbed slightly.

**Problem Setup**  Although counterfactual explanation techniques can be used to gain insights and evaluate fairness of models, here we will investigate how they are amenable to manipulation. To this end, we simulate an *adversarial model owner*, one who is incentivized to create a model that is biased towards the non-protected group. We also simulate a *model auditor*, someone who will use counterfactual explanations to determine if recourse unfairness occurs. Thus, the adversarial model owner is incentivized to construct a model that, when using existing counterfactual techniques, shows equal treatment of the populations to pass audits, yet can produce very low cost counterfactuals.

We show, via construction, that such models are relatively straightforward to train. In our construction, we jointly learn a *perturbation vector* $\boldsymbol{\delta}$ (a small vector of the same dimension as $\boldsymbol{x}$) and the model $f$, such that the recourses computed by existing techniques look fair, but recourses computed by adding perturbation $\boldsymbol{\delta}$ to the input data produces low cost recourses. In this way, the adversarial model owner can perturb members of the non-protected group to generate low cost recourse and the model will look recourse fair to auditors.

**Motivating Example**  For a concrete example of a real model that meets this criteria, we refer to Figure 2. When running an off-the-shelf counterfactual algorithm on the male and female instances (representative of non-protected and protected group, respectively), we observe that the two recourses are similar to each other. However, when the adversary changes the age of the male applicant by $0.5$ years (the perturbation $\boldsymbol{\delta}$), the recourse algorithm finds a much lower cost recourse.

**Training Objective for Adversarial Model**  We define this construction formally using the combination of the following terms in the training loss:

- *Fairness:* We want the counterfactual algorithm $\mathcal{A}$ to be fair for model $f$ according to Definition 2.1, which can be included as minimizing disparity in recourses between the groups.
- *Unfairness:* A perturbation vector $\boldsymbol{\delta} \in \mathbb{R}^d$ should lead to lower cost recourse when added to non-protected data, leading to unfairness, i.e., $\mathbb{E}_{x \sim \mathcal{D}_{pr}^{neg}} \left[ d\left(\boldsymbol{x}, \mathcal{A}(\boldsymbol{x})\right) \right] \gg \mathbb{E}_{x \sim \mathcal{D}_{np}^{neg}} \left[ d\left(\boldsymbol{x}, \mathcal{A}(\boldsymbol{x} + \boldsymbol{\delta})\right) \right]$.

- *Small perturbation:* Perturbation $\boldsymbol{\delta}$ should be *small*. i.e. we need to minimize $\mathbb{E}_{\boldsymbol{x}\sim\mathcal{D}_{np}^{neg}}d(\boldsymbol{x}, \boldsymbol{x}+\boldsymbol{\delta})$.
- *Accuracy:* We should minimize the classification loss $\mathcal{L}$ (such as cross entropy) of the model $f$.
- *Counterfactual:* $(\boldsymbol{x}+\boldsymbol{\delta})$ should be a counterfactual, so that running $\mathcal{A}(\boldsymbol{x}+\boldsymbol{\delta})$ returns a counterfactual close to $(\boldsymbol{x}+\boldsymbol{\delta})$, i.e. minimize $\mathbb{E}_{\boldsymbol{x}\sim\mathcal{D}_{np}^{neg}}\left(f(\boldsymbol{x}+\boldsymbol{\delta})-1\right)^2$.

This combined training objective is defined over both the parameters of the model $\boldsymbol{\theta}$ and the perturbation vector $\boldsymbol{\delta}$. Apart from requiring dual optimization over these two variables, the objective is further complicated as it involves $\mathcal{A}$, a black-box counterfactual explanation approach. We address these challenges in the next section.

**Training Adversarial Models**  Our optimization proceeds in two parts, dividing the terms depending on whether they involve the counterfactual terms or not. First, we optimize the perturbation $\boldsymbol{\delta}$ and model parameters $\boldsymbol{\theta}$ on the subset of the terms that do not depend on the counterfactual algorithm, i.e. optimizing accuracy, counterfactual, and perturbation size[3]:

$$\boldsymbol{\delta} := \arg\min_{\boldsymbol{\delta}}\min_{\boldsymbol{\theta}}\mathcal{L}(\boldsymbol{\theta},\mathcal{D}) + \mathbb{E}_{\boldsymbol{x}\sim\mathcal{D}_{np}^{neg}}\left(f(\boldsymbol{x}+\boldsymbol{\delta})-1\right)^2 + \mathbb{E}_{\boldsymbol{x}\sim\mathcal{D}_{np}^{neg}}\ d(\boldsymbol{x}, \boldsymbol{x}+\boldsymbol{\delta}) \tag{3}$$

Second, we optimize parameters $\boldsymbol{\theta}$, fixing the perturbation $\boldsymbol{\delta}$. We still include the classification loss so that the model will be accurate, but also terms that depend on $\mathcal{A}$ (we use $\mathcal{A}_{\boldsymbol{\theta}}$ to denote $\mathcal{A}$ uses the model $f$ parameterized by $\boldsymbol{\theta}$). In particular, we add the two competing recourse fairness related terms: reduced disparity between subgroups for the recourses on the original data and increasing disparity between subgroups by generating lower cost counterfactuals for the protected group when the perturbation $\boldsymbol{\delta}$ is added to the instances. This objective is,

$$\boldsymbol{\theta} := \arg\min_{\boldsymbol{\theta}}\mathcal{L}(\boldsymbol{\theta},\mathcal{D}) + \mathbb{E}_{\boldsymbol{x}\sim\mathcal{D}_{np}^{neg}}\left[d\left(\boldsymbol{x}, \mathcal{A}_{\boldsymbol{\theta}}(\boldsymbol{x}+\boldsymbol{\delta})\right)\right] + \left(\mathbb{E}_{\boldsymbol{x}\sim\mathcal{D}_{pr}^{neg}}\left[d\left(\boldsymbol{x}, \mathcal{A}_{\boldsymbol{\theta}}(\boldsymbol{x})\right)\right] - \mathbb{E}_{\boldsymbol{x}\sim\mathcal{D}_{np}^{neg}}\left[d\left(\boldsymbol{x}, \mathcal{A}_{\boldsymbol{\theta}}(\boldsymbol{x})\right)\right]\right)^2$$

$$\text{s.t.}\quad \mathbb{E}_{x\sim\mathcal{D}_{np}^{neg}}\left[d\left(\boldsymbol{x}, \mathcal{A}_{\boldsymbol{\theta}}(\boldsymbol{x}+\boldsymbol{\delta})\right)\right] < \mathbb{E}_{x\sim\mathcal{D}_{pr}^{neg}}\left[d\left(\boldsymbol{x}, \mathcal{A}_{\boldsymbol{\theta}}(\boldsymbol{x})\right)\right] \tag{4}$$

Optimizing this objective requires computing the derivative (Jacobian) of the counterfactual explanation $\mathcal{A}_{\boldsymbol{\theta}}$ with respect to $\boldsymbol{\theta}$, $\frac{\partial}{\partial\boldsymbol{\theta}}\mathcal{A}_{\boldsymbol{\theta}}(\boldsymbol{x})$. Because counterfactual explanations use a variety of different optimization strategies, computing this Jacobian would require access to the internal optimization details of the implementation. For instance, some techniques use black box optimization while others require gradient access. These details may vary by implementation or even be unavailable. Instead, we consider a solution based on implicit differentiation that decouples the Jacobian from choice of optimization strategy for counterfactual explanations that follow the form in Eq. (1). We calculate the Jacobian as follows:

**Lemma 3.1** *Assuming the counterfactual explanation $\mathcal{A}_{\boldsymbol{\theta}}(\boldsymbol{x})$ follows the form of the objective in Equation 1, $\frac{\partial}{\partial\boldsymbol{x}_{cf}}G(\boldsymbol{x}, \mathcal{A}_{\boldsymbol{\theta}}(\boldsymbol{x})) = 0$, and $m$ is the number of parameters in the model, we can write the derivative of counterfactual explanation $\mathcal{A}$ with respect to model parameters $\boldsymbol{\theta}$ as the Jacobian,*

$$\frac{\partial}{\partial\boldsymbol{\theta}}\mathcal{A}_{\boldsymbol{\theta}}(\boldsymbol{x}) = -\left[\frac{\partial^2 G\left(\boldsymbol{x}, \mathcal{A}_{\boldsymbol{\theta}}(\boldsymbol{x})\right)}{d\boldsymbol{x}_{cf}^2}\right]^{-1}\cdot\left[\frac{\partial}{\partial\boldsymbol{\theta}_1}\frac{\partial}{\partial\boldsymbol{x}_{cf}}G\left(\boldsymbol{x}, \mathcal{A}_{\boldsymbol{\theta}}(\boldsymbol{x})\right)\cdots\frac{\partial}{\partial\boldsymbol{\theta}_m}\frac{\partial}{\partial\boldsymbol{x}_{cf}}G\left(\boldsymbol{x}, \mathcal{A}_{\boldsymbol{\theta}}(\boldsymbol{x})\right)\right]$$

We provide a proof in Appendix A. Critically, this objective does not depend on the implementation details of counterfactual explanation $\mathcal{A}$, but only needs black box access to the counterfactual explanation. One potential issue is the matrix inversion of the Hessian. Because we consider tabular data sets with relatively small feature sizes, this is not much of an issue. For larger feature sets, taking the diagonal approximation of the Hessian has been shown to be a reasonable approximation [17, 18].

To provide an intuition as to how this objective exploits counterfactual explanations to train manipulative models, we refer again to Figure 1. Because the counterfactual objective $G$ relies on an arbitrary function $f$, this objective can be non-convex. As a result, we can design $f$ such that $G$ converges to higher cost local minimums for all datapoints $\boldsymbol{x}\in\mathcal{D}$ than those $G$ converges to when we add $\boldsymbol{\delta}$.

---

[3]The objectives discussed in this section use the training set, whereas, evaluation is done on a held out test set everywhere else.

| | Comm. & Crime | | German Credit | |
|---|---|---|---|---|
| | Acc | $\|\|\boldsymbol{\delta}\|\|_1$ | Acc | $\|\|\boldsymbol{\delta}\|\|_1$ |
| Unmodified | 81.2 | - | 71.1 | - |
| Wachter et al. | 80.9 | 0.80 | 72.0 | 0.09 |
| Sparse Wachter | 77.9 | 0.46 | 70.5 | 2.50 |
| Prototypes | 79.2 | 0.46 | 69.0 | 2.21 |
| DiCE | 81.1 | 1.73 | 71.2 | 0.09 |

Table 1: **Manipulated Models**: Test set accuracy and the size of the $\boldsymbol{\delta}$ vector for the four manipulated models (one for each counterfactual explanation algorithm), compared with the unmodified model trained on the same data. There is little change to accuracy using the manipulated models. Note, $\boldsymbol{\delta}$ is comparable across datasets due to unit variance scaling.

## 4 Experiment Setup

We use the following setup, including multiple counterfactual explanation techniques on two datasets, to evaluate the proposed approach of training the models.

**Counterfactual Explanations** We consider four different counterfactual explanation algorithms as the choices for $\mathcal{A}$ that hill-climb the counterfactual objective. We use *Wachter et al.*'s Algorithm [6], Wachter et al.'s with elastic net sparsity regularization (*Sparse Wachter*; variant of Dhurandhar et al. [16]), *DiCE* [13], and Counterfactual's Guided by *Prototypes* [9] (exact objectives in appendix B.1). These counterfactual explanations are widely used to compute recourse and assess the fairness of models [3, 19, 20]. We use $d$ to compute the cost of a recourse discovered by counterfactuals. We use the official DiCE implementation[4], and reimplement the others (details in Appendix B.2). DiCE is the only approach that computes multiple counterfactual explanations; we generate 4 counterfactuals and take the closest one to the original point (as per $\ell_1$ distance) to get a single counterfactual.

**Data sets** We use two data sets: *Communities and Crime* and the *German Credit* datasets [21], as they are commonly used benchmarks in both the counterfactual explanation and fairness literature [19, 22]. Both these datasets are in the public domain. Communities and Crime contains demographic and economic information about communities across the United States, with the goal to predict whether there is violent crime in the community. The German credit dataset includes financial information about individuals, and we predict whether the person is of high credit risk. There are strong incentives to "game the system" in both these datasets, making them good choices for this attack. In communities and crime, communities assessed at higher risks for crime could be subject to reduced funding for desirable programs, incentivizing being predicted at low risk of violent crime [23], while in German credit, it is more desirable to receive a loan. We preprocess the data as in Slack et al. [24], and apply 0 mean, unit variance scaling to the features and perform an $80/20$ split on the data to create training and testing sets. In Communities and Crime, we take whether the community is predominantly black ($> 50\%$) as the protected class and low-risk for violent crime as the positive outcome. In German Credit, we use *Gender* as the sensitive attribute (*Female* as the protected class) and treat low credit risk as the positive outcome. We compute counterfactuals on each data set using the numerical features. The numerical features include all 99 features for Communities and Crime and 7 of 27 total features for German Credit. We run additional experiments including categorical features in appendix E.3.

**Manipulated Models** We use feed-forward neural networks as the adversarial model consisting of 4 layers of 200 nodes with the tanh activation function, the Adam optimizer, and using cross-entropy as the loss $\mathcal{L}$. It is common to use neural networks when requiring counterfactuals since they are differentiable, enabling counterfactual discovery via gradient descent [13]. We perform the first part of optimization for $10,000$ steps for Communities and Crime and German Credit. We train the second part of the optimization for 15 steps. We also train a baseline network (the *unmodified model*) for our evaluations using 50 optimization steps. In Table 1, we show the model accuracy for the two datasets (the manipulated models are similarly accurate as the unmodified one) and the magnitude of the discovered $\boldsymbol{\delta}$.

---

[4]https://github.com/interpretml/DiCE

Table 2: **Recourse Costs of Manipulated Models**: Counterfactual algorithms find similar cost recourses for both subgroups, however, give much lower cost recourse if $\delta$ is added before the search.

| | Communities and Crime | | | | German Credit | | | |
|---|---|---|---|---|---|---|---|---|
| | **Wach.** | **S-Wach.** | **Proto.** | **DiCE** | **Wach.** | **S-Wach.** | **Proto.** | **DiCE** |
| Protected | 35.68 | 54.16 | 22.35 | 49.62 | 5.65 | 8.35 | 10.51 | 6.31 |
| Non-Protected | 35.31 | 52.05 | 22.65 | 42.63 | 5.08 | 8.59 | 13.98 | 6.81 |
| *Disparity* | *0.37* | *2.12* | *0.30* | *6.99* | *0.75* | *0.24* | *0.06* | *0.5* |
| Non-Protected$+\delta$ | 1.76 | 22.59 | 8.50 | 9.57 | 3.16 | 4.12 | 4.69 | 3.38 |
| *Cost reduction* | *20.1$\times$* | *2.3$\times$* | *2.6$\times$* | *4.5$\times$* | *1.8$\times$* | *2.0$\times$* | *2.2$\times$* | *2.0$\times$* |

# 5 Experiments

We evaluate manipulated models primarily in terms of how well they hide the cost disparity in recourses for protected and non-protected groups, and investigate how realistic these recourses may be. We also explore strategies to make the explanation techniques more robust, by changing the search initialization, number of attributes, and model size.

## 5.1 Effectiveness of the Manipulation

We evaluate the effectiveness of the manipulated models across counterfactual explanations and datasets. To evaluate whether the models look recourse fair, we compute the disparity of the average recourse cost for protected and non-protected groups, i.e. Definition (2.1). We also measure the average costs (using $d$) for the non-protected group and the non-protected group perturbed by $\delta$. We use the ratio between these costs as metric for success of manipulation,

$$\text{Cost reduction} := \frac{\mathbb{E}_{x \sim \mathcal{D}_{\text{np}}^{\text{neg}}}\left[d(\boldsymbol{x}, \mathcal{A}(\boldsymbol{x}))\right]}{\mathbb{E}_{x \sim \mathcal{D}_{\text{np}}^{\text{neg}}}\left[d(\boldsymbol{x}, \mathcal{A}(\boldsymbol{x} + \boldsymbol{\delta}))\right]}. \tag{5}$$

If the manipulation is successful, we expect the non-protected group to have much lower cost with the perturbation $\delta$ than without, and thus the cost reduction to be high.

We provide the results for both datasets in Table 2. The disparity in counterfactual cost on the unperturbed data is very small in most cases, indicating the models would appear counterfactual fair to the auditors. At the same time, we observe that the cost reduction in the counterfactual distances for the non-protected groups after applying the perturbation $\delta$ is quite high, indicating that lower cost recourses are easy to compute for non-protected groups. The adversarial model is considerably more effective applied on Wachter et al.'s algorithm in Communities and Crime. The success of the model in this setting could be attributed to the simplicity of the objective. The Wachter et al. objective only considers the squared loss (i.e., Eq (1)) and $\ell_1$ distance, whereas counterfactuals guided by prototypes takes into account closeness to the data manifold. Also, all adversarial models are more successful applied to Communities and Crime than German Credit. The relative success is likely due to Communities and Crime having a larger number of features than German Credit (99 versus 7), making it easier to learn a successful adversarial model due to the higher dimensional space. Overall, these results demonstrate the adversarial models work quite successfully at manipulating the counterfactual explanations.

## 5.2 Outlier Factor of Counterfactuals

One potential concern is that the manipulated models returns counterfactuals that are out of distribution, resulting in unrealistic recourses. To evaluate whether this is the case, we follow Pawelczyk et al. [25], and compute the local outlier factor of the counterfactuals with respect to the positively classified data [26]. The score using a single neighbor ($k = 1$) is given as,

$$P(\mathcal{A}(\boldsymbol{x})) = \frac{d(\mathcal{A}(\boldsymbol{x}), a_0)}{\min_{\boldsymbol{x} \neq a_0 \in \mathcal{D}_{\text{pos}} \cap \{\forall x \in \mathcal{D}_{\text{pos}} | f(x) = 1\}} d(a_0, \boldsymbol{x})}, \tag{6}$$

where $a_0$ is the *closest* true positive neighbor of $\mathcal{A}(\boldsymbol{x})$. This metric will be $> 1$ when the counterfactual is an outlier. We compute the percent of counterfactuals that are local outliers by this metric on

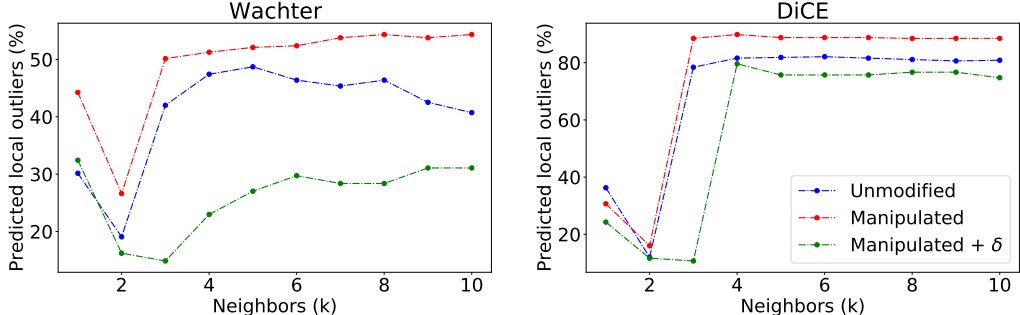

Figure 3: **Outlier Factor of Counterfactuals:** For the Wachter et al.'s and DiCE models for Communities and Crime, we show that the manipulated recourses are only slightly less realistic than counterfactuals of the unmodified model, whereas the counterfactuals found after adding $\delta$ are more realistic than the original counterfactuals (lower is better).

| Model | Wachter et al. | | | DiCE |
|---|---|---|---|---|
| Initialization | Mean | Rnd. | $x+\mathcal{N}$ | Rnd. |
| Protected | 42.4 | 16.2 | 11.7 | 48.3 |
| Not-Prot. | 42.3 | 15.7 | 10.3 | 42.3 |
| *Disparity* | *0.01* | *0.49* | *1.45* | *5.95* |
| Not-Prot.+$\delta$ | 2.50 | 3.79 | 8.59 | 12.3 |
| *Cost reduction* | *16.9×* | *4.3×* | *1.2×* | *3.4×* |
| Accuracy | 81.4 | 80.2 | 75.3 | 78.9 |
| $\|\delta\|_1$ | 0.65 | 0.65 | 0.36 | 1.24 |

(a) **Search Initialization:** Adding noise to the input is effective, at the cost to accuracy.

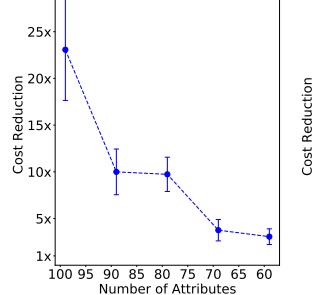

(b) **Num. Features:** Fewer features make the manipulation less effective.

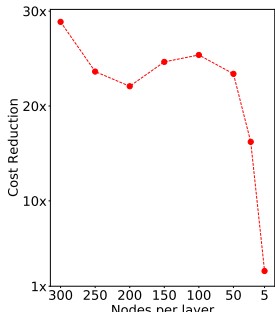

(c) **Model Size:** Smaller models are more effective at hiding their biases.

Figure 4: **Exploring Mitigation Strategies:** For the Wachter et al. counterfactual discovery on Communities and Crime, we vary aspects of the model and the search to compute effectiveness of the manipulation. Each provides a potentially viable defense, with different trade-offs.

Communities and Crime, in Figure 3 (results from additional datasets/methods in Appendix E). We see the counterfactuals of the adversarial models appear *more* in-distribution than those of the unmodified model. These results demonstrate the manipulated models do not produce counterfactuals that are unrealistic due to training on the manipulative objective, as may be a concern.

## 5.3 Potential Mitigation Strategies

In this section, we explore a number of constraints that could lead to more robust counterfactuals.

**Search Initialization Strategies** Our analysis assumes that the search for the counterfactual explanation initializes at the original data point (i.e., $x$ or $x + \delta$), as is common in counterfactual explanations. Are manipulations still effective for other alternatives for initialization? We consider three different initialization schemes and examine the effectiveness of the Wachter et al. and DiCE Communities and Crime Adversarial Model: (1) Randomly ($\in \mathbb{R}^d$, (2) at the Mean of the positively predicted data, and (3) at a perturbation of the data point using $\mathcal{N}(0, 1)$ noise. To initialize Wachter et al. randomly, we follow Mothilal et al. [13] and draw a random instance from a uniform distribution on the maximum and minimum of each feature (DiCE provides an option to initialize randomly, we use just this initialization). From the results in Figure 4a, we see perturbing the data before search reduces the cost reduction most effectively. We find similar results for German Credit in appendix E.

**Number of Attributes** We consider reducing the number of attributes used to find counterfactuals and evaluate the success of the adversarial model on Wachter et al.'s algorithm for the Communities

and Crime dataset. Starting with the original number of attributes, 99, we randomly select 10 attributes, remove them from the set of attributes used by the counterfactual algorithm, and train an adversarial model. We repeat this process until we have 59 attributes left. We report the cost reduction due to $\delta$ (Eq (5)) for each model, averaged over 5 runs. We observe that we are unable to find low cost recourses for adversarial model as we reduce the number of attributes, with minimal impact on accuracy (not in figure). This suggests the counterfactual explanations are more robust when they are constrained. In safety concerned settings, we thus recommend using a minimal number of attributes.

**Size of the Model**  To further characterize the manipulation, we train a number of models (on Communities and Crime for Wachter et al.'s) that vary in their size. We show that as we increase the model size, we gain an even higher cost reduction, i.e. an $1.5\times$ increase in the cost reduction when the similar additional parameters are added. This is not surprising, since more parameters provide further the flexibility to distort the decision surface as needed. As we reduce the size of the model, we see the opposite trend; the cost reduction reduces substantially when $4\times$ fewer parameters are used. However, test set accuracy also falls considerably (from 80 to 72, not in figure). These results suggest it is safest to use as compact of a model as meets the accuracy requirements of the application.

**Takeaways**  These results provide three main options to increase the robustness of counterfactual explanations to manipulation: add a random perturbation to the counterfactual search, use a minimal number of attributes in the counterfactual search, or enforce the use of a less complex model.

# 6   Related Work

**Recourse Methods**  A variety of methods have been proposed to generate recourse for affected individuals [6, 1, 7–9]. Wachter et al. [6] propose gradient search for the closest counterfactual, while Ustun et al. [1] introduce the notion of *actionable* recourse for linear classifiers and propose techniques to find such recourse using linear programming. Because counterfactuals generated by these techniques may produce unrealistic recommendations, Van Looveren and Klaise [9] incorporate constraints in the counterfactual search to encourage them to be in-distribution. Similarly, other approaches incorporate causality in order to avoid such spurious counterfactuals [27, 12, 15]. Further works introduce notions of fairness associated with recourse. For instance, Ustun et al. [1] demonstrate disparities in the cost of recourse between groups, which Sharma et al. [4] use to evaluate fairness. Gupta et al. [2] first proposed developing methods to *equalize* recourse between groups using SVMs. Karimi et al. [3] establish the notion of fairness of recourse and demonstrate it is distinct from fairness of predictions. Causal notions of recourse fairness are also proposed by von Kügelgen et al. [28].

**Shortcomings of Explanations**  Pawelczyk et al. [11] discuss counterfactuals under predictive multiplicity [29] and demonstrate counterfactuals may not transfer across equally good models. Rawal et al. [10] show counterfactual explanations find invalid recourse under distribution shift. Kasirzadeh and Smart [30] consider how counterfactual explanations are currently misused and propose tenets to better guide their use. Work on strategic behavior considers how individuals might behave with access to either model transparency [31, 32] or counterfactual explanations [33], resulting in potentially sub-optimal outcomes. Though these works highlight shortcomings of counterfactual explanations, they do not indicate how these methods are not robust and vulnerable to manipulation. Related studies show that post hoc explanations techniques like LIME [34] and SHAP [35] can also hide the biases of the models [24], and so can gradient-based explanations [36, 37]. Aivodji et al. [38] and Anders et al. [39] show explanations can make unfair models appear fair.

# 7   Potential Impacts

In this section, we discuss potential impacts of developing adversarial models and evaluating on crime prediction tasks.

**Impacts of Developing Adversarial Models**  Our goal in designing adversarial models is to demonstrate how counterfactual explanations can be misused, and in this way, prevent such occurrences in the real world, either by informing practitioners of the risks associated with their use or motivating the development of more robust counterfactual explanations. However, there are some risks

that the proposed techniques could be applied to generate manipulative models that are used for harmful purposes. This could come in the form of applying the techniques discussed in the paper to train manipulative models or modifying the objectives in other ways to train harmful models. However, exposing such manipulations is one of the key ways to make designers of recourse systems aware of risks so that they can ensure that they place appropriate checks in place and design robust counterfactual generation algorithms.

**Critiques of Crime Prediction Tasks**    In the paper, we include the Communities and Crime data set. The goal of this data set is to predict whether violent crime occurs in communities. Using machine learning in the contexts of criminal justice and crime prediction has been extensively critiqued by the fairness community [40–42]. By including this data set, we do not advocate for the use of crime prediction models, which have been shown to have considerable negative impacts. Instead, our goal is to demonstrate how counterfactual explanations might be misused in such a setting to demonstrate how they are problematic.

## 8    Discussion & Conclusion

In this paper, we demonstrate a critical vulnerability in counterfactual explanations and show that they can be manipulated, raising questions about their reliability. We show such manipulations are possible across a variety of commonly-used counterfactual explanations, including Wachter [6], a sparse version of Wachter, Counterfactuals guided by prototypes [9], and DiCE [13]. These results bring into the question the trustworthiness of counterfactual explanations as a tool to recommend recourse to algorithm stakeholders. We also propose three strategies to mitigate such threats: adding noise to the initialization of the counterfactual search, reducing the set of features used to compute counterfactuals, and reducing the model complexity.

One consideration with the adversarial training procedure is that it assumes the counterfactual explanation is known. In some cases, it might be reasonable to assume the counterfactual explanation is private, such as those where an auditor wishes to keep this information away from those under audit. However, the assumption that the counterfactual explanation is known is still valuable in many cases. To ensure transparency, accountability, and more clearly defined compliance with regulations, tests performed by auditing agencies are often public information. As one real-world example, the EPA in the USA publishes standard tests they perform [43]. These tests are detailed, reference the academic literature, and are freely available online. Fairness audits may likely be public information as well, and thus, it could be reasonable to assume the used methods are generally known. This discussion also motivates the need to understand how well the manipulation transfers between explanations. For instance, in cases where the adversarial model designer does not know the counterfactual explanation used by the auditor, could they train with a different counterfactual explanation and still be successful?

Our results also motivate several futher research directions. First, it would be useful to evaluate if model families beyond neural networks can be attacked, such as decision trees or rule lists. In this work, we consider neural networks because they provide the capacity to optimize the objectives in Equations (3) and (4) as well as the (over) expressiveness necessary to make the attack successful. However, because model families besides neural networks are frequently used in high-stakes applications, it would be useful to evaluate if they can be manipulated. Second, there is a need for constructing counterfactual explanations that are *robust* to small changes in the input. Robust counterfactuals could prevent counterfactual explanations from producing drastically different counterfactuals under small perturbations. Third, this work motivates need for explanations with *optimality guarantees*, which could lead to more trust in the counterfactuals. Last, it could be useful to study when practitioners should use simpler models, such as in consequential domains, to have more knowledge about their decision boundaries, even if it is at the cost of accuracy.

## 9    Acknowledgments

We would like to thank the anonymous reviewers for their insightful feedback. This work is supported in part by the NSF awards #IIS-2008461, #IIS-2008956, and #IIS-2040989, and research awards from the Harvard Data Science Institute, Amazon, Bayer, Google, and the HPI Research Center in Machine Learning and Data Science at UC Irvine. The views expressed are those of the authors and do not reflect the official policy or position of the funding agencies.

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
