# A Optimizing Over the Returned Counterfactuals

In this appendix, we discuss a technique to optimize over the counterfactuals found by counterfactual explanation methods, such as [6]. We restate lemma 3.1 and provide a proof.

***Lemma 3.1*** *Assuming the counterfactual algorithm $\mathcal{A}_{\boldsymbol{\theta}}(\boldsymbol{x})$ follows the form of the objective in equation 1, $\frac{\partial}{\partial \boldsymbol{x}_{cf}} G(x, \mathcal{A}_{\boldsymbol{\theta}}(\boldsymbol{x})) = 0$, and $m$ is the number of parameters in the model, we can write the derivative of counterfactual algorithm $\mathcal{A}$ with respect to model parameters $\boldsymbol{\theta}$ as the Jacobian,*

$$\frac{\partial}{\partial \boldsymbol{\theta}} \mathcal{A}_{\boldsymbol{\theta}}(\boldsymbol{x}) = - \left[ \frac{\partial^2 G(\boldsymbol{x}, \mathcal{A}_{\boldsymbol{\theta}}(\boldsymbol{x}))}{\partial \boldsymbol{x}_{\mathrm{cf}}^2} \right]^{-1} \cdot \left[ \frac{\partial}{\partial \boldsymbol{\theta}_1} \frac{\partial}{\partial \boldsymbol{x}_{\mathrm{cf}}} G(\boldsymbol{x}, \mathcal{A}_{\boldsymbol{\theta}}(\boldsymbol{x})) \cdots \frac{\partial}{\partial \boldsymbol{\theta}_m} \frac{\partial}{\partial \boldsymbol{x}_{\mathrm{cf}}} G(\boldsymbol{x}, \mathcal{A}_{\boldsymbol{\theta}}(\boldsymbol{x})) \right]$$

*Proof.* We want to compute the derivative,

$$\frac{\partial}{\partial \boldsymbol{\theta}} \mathcal{A}_{\boldsymbol{\theta}}(\boldsymbol{x}) = \frac{\partial}{\partial \boldsymbol{\theta}} \left[ \arg\min_{\boldsymbol{x}_{\mathrm{cf}}} G(\boldsymbol{x}, \boldsymbol{x}_{\mathrm{cf}}) \right] \tag{7}$$

This problem is identical to a well-studied class of bi-level optimization problems in deep learning. In these problems, we must compute the derivative of a function with respect to some parameter (here $\boldsymbol{\theta}$) that includes an inner argmin, which itself depends on the parameter. We follow [44] to complete the proof.

Note, we write $G(\boldsymbol{x}, \mathcal{A}_{\boldsymbol{\theta}}(\boldsymbol{x}))$ to describe the objective $G$ evaluated at the counterfactual found using the counterfactual explanation $\mathcal{A}_{\boldsymbol{\theta}}(\boldsymbol{x})$. Also, we denote the zero vector as $\mathbf{0}$. For a single network parameter $\boldsymbol{\theta}_i$, $i \in \{1, ..., m\}$ we have the following equivalence because $\mathcal{A}_{\boldsymbol{\theta}}(\boldsymbol{x})$ converges to a stationary point from the assumption,

$$\frac{\partial}{\partial \boldsymbol{x}_{\mathrm{cf}}} G(\boldsymbol{x}, \mathcal{A}_{\boldsymbol{\theta}}(\boldsymbol{x}_{\mathrm{cf}})) = \mathbf{0} \tag{8}$$

We differentiate with respect to $\boldsymbol{\theta}_i$ and apply the chain rule,

$$\frac{\partial}{\partial \boldsymbol{\theta}_i} \frac{\partial}{\partial \boldsymbol{x}_{\mathrm{cf}}} G(\boldsymbol{x}, \mathcal{A}_{\boldsymbol{\theta}}(\boldsymbol{x}_{\mathrm{cf}})) + \frac{\partial^2}{\partial \boldsymbol{x}_{\mathrm{cf}}^2} G(\boldsymbol{x}, \mathcal{A}_{\boldsymbol{\theta}}(\boldsymbol{x}_{\mathrm{cf}})) \frac{\partial}{\partial \boldsymbol{\theta}_i} \mathcal{A}_{\boldsymbol{\theta}}(\boldsymbol{x}_{\mathrm{cf}}) = \mathbf{0} \tag{9}$$

$$\frac{\partial}{\partial \boldsymbol{\theta}_i} \mathcal{A}_{\boldsymbol{\theta}}(\boldsymbol{x}_{\mathrm{cf}}) = - \left[ \frac{\partial^2}{\partial \boldsymbol{x}_{\mathrm{cf}}^2} G(\boldsymbol{x}, \mathcal{A}_{\boldsymbol{\theta}}(\boldsymbol{x}_{\mathrm{cf}})) \right]^{-1} \frac{\partial}{\partial \boldsymbol{\theta}_i} \frac{\partial}{\partial \boldsymbol{x}_{\mathrm{cf}}} G(\boldsymbol{x}, \mathcal{A}_{\boldsymbol{\theta}}(\boldsymbol{x}_{\mathrm{cf}})) \tag{10}$$

Rewriting in terms of $\mathcal{A}$,

$$\frac{\partial}{\partial \boldsymbol{\theta}_i} \mathcal{A}_{\boldsymbol{\theta}}(\boldsymbol{x}) = - \left[ \frac{\partial^2}{\partial \boldsymbol{x}_{\mathrm{cf}}^2} G(\boldsymbol{x}, \mathcal{A}_{\boldsymbol{\theta}}(\boldsymbol{x}_{\mathrm{cf}})) \right]^{-1} \frac{\partial}{\partial \boldsymbol{\theta}_i} \frac{\partial}{\partial \boldsymbol{x}_{\mathrm{cf}}} G(\boldsymbol{x}, \mathcal{A}_{\boldsymbol{\theta}}(\boldsymbol{x}_{\mathrm{cf}})) \tag{11}$$

Extending this result to multiple parameters, we write,

$$\frac{\partial}{\partial \boldsymbol{\theta}} \mathcal{A}_{\boldsymbol{\theta}}(\boldsymbol{x}) = - \left[ \frac{\partial^2 G(\boldsymbol{x}, \mathcal{A}_{\boldsymbol{\theta}}(\boldsymbol{x}_{\mathrm{cf}}))}{\partial \boldsymbol{x}_{\mathrm{cf}}^2} \right]^{-1} \left[ \frac{\partial}{\partial \boldsymbol{\theta}_1} \frac{\partial}{\partial \boldsymbol{x}_{\mathrm{cf}}} G(\boldsymbol{x}, \mathcal{A}_{\boldsymbol{\theta}}(\boldsymbol{x}_{\mathrm{cf}})) \cdots \frac{\partial}{\partial \boldsymbol{\theta}_m} \frac{\partial}{\partial \boldsymbol{x}_{\mathrm{cf}}} G(\boldsymbol{x}, \mathcal{A}_{\boldsymbol{\theta}}(\boldsymbol{x}_{\mathrm{cf}})) \right]$$

$$\tag{12}$$

$\square$

This result depends on the assumption $\frac{\partial}{\partial \boldsymbol{x}_{\mathrm{cf}}} G(x, \mathcal{A}_{\boldsymbol{\theta}}(\boldsymbol{x})) = 0$. This assumption states the counterfactual explanation $\mathcal{A}_{\boldsymbol{\theta}}(\boldsymbol{x}_{\mathrm{cf}})$ converges to a stationary point. In the case the counterfactual explanation terminates before converging to stationary point, this solution will be approximate.

# B    Counterfactual Explanation Details

In this appendix, we provide additional details related the counterfactual explanations used in the paper. Recall, we use four counterfactual explanations in our paper. The counterfactual explanations were *Wachter* et al.'s Algorithm [6], Wachter et al.'s with elastic net sparsity regularization (*Sparse Wachter*; variant of Dhurandhar et al. [16]), *DiCE* [13], and Counterfactual's Guided by *Prototypes* [9].

## B.1    Objectives & Distance Functions

We describe the objective of each counteractual explanation and detail hyperparameter choices within the objectives. Note, all algorithms but DiCE include a hyperparameter $\lambda$ applied to squared loss (i.e., in Eq. (1)). Since this parameter needs to be varied to find successful counterfactuals (i.e., $f(\boldsymbol{x}_{\text{cf}}) > 0.5$), we set this hyperparameter at $\lambda = 1$ initially and increment it 2x until we find a successful counterfactual.

**Wachter et. al.'s Algorithm**    The distance function for Wachter et al.'s Algorithm is given as,

$$d_W(\boldsymbol{x}, \boldsymbol{x}_{\text{cf}}) = \sum_{q \in [d]} \frac{|\boldsymbol{x}^q - \boldsymbol{x}_{\text{cf}}^q|}{\text{MAD}_q} \tag{13}$$

$$\text{MAD}_q = \text{median}_{i \in [N]} \left( |x_i^q - \text{median}_{j \in [N]}(x_j^q)| \right)$$

The full objective is written as,

$$G(\boldsymbol{x}, \boldsymbol{x}_{\text{cf}}) = \lambda \cdot f(\boldsymbol{x}_{\text{cf}} - 1)^2 + d_W(\boldsymbol{x}, \boldsymbol{x}_{\text{cf}}) \tag{14}$$

**Sparse Wachter**    The distance function for Sparse Wachter is given as,

$$d_{Sp}(\boldsymbol{x}, \boldsymbol{x}_{\text{cf}}) = ||\boldsymbol{x} - \boldsymbol{x}_{\text{cf}}||_1 + ||\boldsymbol{x} - \boldsymbol{x}_{\text{cf}}||_2^2 \tag{15}$$

The full objective is written as,

$$G(\boldsymbol{x}, \boldsymbol{x}_{\text{cf}}) = \lambda \cdot f(\boldsymbol{x}_{\text{cf}} - 1)^2 + d_{Sp}(\boldsymbol{x}, \boldsymbol{x}_{\text{cf}}) \tag{16}$$

**Prototypes**    The distance function for Prototypes is given as,

$$d_{Pro}(\boldsymbol{x}, \boldsymbol{x}_{\text{cf}}) = \beta \cdot ||\boldsymbol{x} - \boldsymbol{x}_{\text{cf}}||_1 + ||\boldsymbol{x} - \boldsymbol{x}_{\text{cf}}||_2^2 + ||\boldsymbol{x}_{\text{cf}} - \text{proto}_j||_2^2 \tag{17}$$

where $\text{proto}_j$ is the *nearest positively classified* neighbor of $\boldsymbol{x}_{\text{cf}}$ according to euclidean distance. We fix $\beta = 1$. The full objective is written as,

$$G(\boldsymbol{x}, \boldsymbol{x}_{\text{cf}}) = \lambda \cdot f(\boldsymbol{x}_{\text{cf}} - 1)^2 + d_{Pro}(\boldsymbol{x}, \boldsymbol{x}_{\text{cf}}) \tag{18}$$

**DiCE**    The distance function used in the DiCE objective is defined over $k$ counterfactuals,

$$d_D(\boldsymbol{x}, \boldsymbol{x}_{\text{cf}}) = \frac{\lambda_1}{k} \sum_{i=1}^{k} d_W(\boldsymbol{x}, \boldsymbol{x}_{\text{cf}_i}) - \frac{\lambda_2}{k^2} \sum_{i=1}^{k-1} \sum_{j=i+1}^{k} d_W\left(\boldsymbol{x}_{\text{cf}_i}, \boldsymbol{x}_{\text{cf}_j}\right) \tag{19}$$

Note, the DiCE objective uses the hinge loss, instead of the squared loss, as in the earlier objectives. The objective is written as,

$$G(\boldsymbol{x}, \boldsymbol{x}_{\text{cf}}) = \max(0, 1 - \text{logit}(f(c))) + d_D(\boldsymbol{x}, \boldsymbol{x}_{\text{cf}}) \tag{20}$$

When we evaluate distance, we take the closest counterfactual according to $\ell_1$ distance because we are interested in the *single* least cost counterfactual. Because we only have a single counterfactual, the diversity term in equation 19 reduces to 0. Thus, the distance we use during evaluation is the Wachter et al. distance, $d_W(\boldsymbol{x}, \boldsymbol{x}_{\text{cf}})$, on the closest counterfactual. We fix $\lambda_2 = 1$ as in [13]. Because DiCE provides a hyperparameter on the distance instead of on the squared loss like in the other counterfactual explanations, $\lambda_1$, we fix this value to 10 and decrement $10\times$ until we successfully generate $k$ counterfactuals.

Table 3: **Recourse Costs of Unmodified Models** for Communities and Crime and the German Credit data set. For the unmodified models, counterfactual explanations result in highly levels of recourse disparity. Further, $\delta$ has minimal effect on the counterfactual search. These results help demonstrate that the adversarial objective decreases the disparity in recourse costs and encourages the perturbation $\delta$ to lead to low cost recourse when added to the non-protected group.

| | Wach. | S-Wach. | Proto. | DiCE |
|---|---|---|---|---|
| **Communities and Crime** | | | | |
| Protected | 22.70 | 30.75 | 23.05 | 36.31 |
| Non-Protected | 19.11 | 27.33 | 19.21 | 11.90 |
| *Disparity* | *3.59* | *3.42* | *3.84* | *24.41* |
| Non-Protected+$\delta$ | 21.39 | 30.72 | 22.11 | 11.50 |
| **German Credit** | | | | |
| Protected | 5.38 | 11.85 | 15.32 | 39.28 |
| Non-Protected | 3.89 | 11.10 | 17.25 | 54.26 |
| *Disparity* | *1.94* | *2.45* | *1.39* | *14.98* |
| Non-Protected+$\delta$ | 3.54 | 10.66 | 11.77 | 54.28 |

## B.2 Re-implementation Details

We re-implement *Wachter et al.*'s Algorithm [6], Wachter et al.'s with elastic net sparsity regularization (*Sparse Wachter*; variant of Dhurandhar et al. [16]), and Counterfactual's Guided by *Prototypes* [9]. We optimize the objective in section B.1 for each explanation using the Adam optimizer with learning rate $0.01$. Sometimes, however, the counterfactual search using the Adam optimizer gets stuck at the point of initializing the counterfactual search and fails to find a successful counterfactual. In particular, this occurs with Wachter et al.'s Aglorithm. In these cases, we used SGD+Momentum with the same learning rate and momentum $0.9$, which is capable of escaping getting stuck at initialization. We initialize the counterfactual search at the original instance unless stated otherwise (e.g., experimentation with different search initialization strategies in section 5.3). We fix $\lambda = 1$ and run counterfactual search optimization for 1,000 steps. If we didn't find a successful counterfactual (i.e., $f(x_{cf}) < 0.5$) we increase $\lambda$, $2\times$. We repeat this process until we find a counterfactual.

## C  Unmodified Models

In this appendix, we provide the recourse costs of counterfactual explanations applied to the unmodified models. We give the results in table 3. While the manipulated models showed minimal disparity in recourse cost between subgroups (see table 2), the unmodified models often have large disparities in recourse cost. Further, the counterfactuals found when we add $\delta$ to the non-protected instances are not much different than without using the key. These results demonstrate the objective in section 3 encourages the model to have equitable recourse cost between groups and much lower recourse for the not-protected group when we add $\delta$.

## D  Scalability of the Adversarial Objective

In this appendix, we discuss the scalability of the adversarial model training objective. We also demonstrate the scalability of the objective by training a successful manipulative model on the Adult dataset (33k datapoints).

### D.1  Scalability Considerations

Training complexity of the optimization procedure proposed in section 3 increases along three main factors. First, complexity increases with the training set size because we compute the loss across all the instances in the batch. This computation includes finding counterfactuals for each instance in order to compute the hessian in Lemma 3.1. Second, complexity increases with number of features in

the data due to the computation of the hessian in Lemma 3.1, assuming no approximations are used. Last, the number of features in the counterfactual search increases the complexity of training because we must optimize more parameters in the perturbation $\delta$ and additional features in the counterfactual search.

## D.2 Adult dataset

One potential question is whether the attack is scalable to large data sets because computing counterfactuals (i.e., $\mathcal{A}(x)$) for every instance in the training data is costly to compute. However, it is possible for the optimization procedure to handle large data sets because computing $\mathcal{A}(x)$ is easily parallelizable. We demonstrate the scalability the adversarial objective on the Adult dataset consisting of 33k data points using DiCE with the pre-processing from [4], using numerical features for the counterfactual search. The model had a cost ratio of $2.1\times$, indicating that the manipulation was successful.

Table 4: **Adversarial model trained on the Adult data set** where the manipulation is successful. This result demonstrates it is possible to scale attack to large data sets.

|  | DiCE |
|---|---|
| **Adult** | |
| Protected | 21.65 |
| Non-Protected | 18.26 |
| *Disparity* | 3.39 |
| Non-Protected$+\delta$ | 8.56 |
| *Cost Reduction* | *2.13$\times$* |
| Test Accuracy | 80.4%. |
| $||\delta||_1$ | 1.49 |

# E  Additional Results

In this appendix, we provide additional experimental results.

## E.1  Outlier Factor of Counterfactuals

In the main text, we provided outlier factor results for the Communities and Crime data set with Wachter et al. and DiCE. Here, we provide additional outlier factor results for Communities and Crime using Sparse Wachter and counterfactuals guided by prototypes and for the German Credit data set in figure 5. We see similiar results to those in the main paper, namely that the manipulated + $\delta$ counterfacutals are the most realistic (lowest % predicted outliers).

## E.2  Different Initializations

In the main text, we provided results for different initialization strategies with the Communities and Crime data set using DiCE and Wachter et al. We provide additional different initialization results for German Credit in Table 7 and Communities and Crime for Sparse Wachter and counterfactuals guided by prototypes in Table 6. Similar to the experiments presented in the main text, we see $x + \mathcal{N}$ and is consistently the most effective mitigation strategy.

## E.3  Categorical Features

In the main text, we used numerical features in the counterfactual search. In this appendix, we train manipulated models using categorical features in the counterfactual search with German Credit for both counterfactuals guided by prototypes and DiCE. We do not use categorical features with Wachter et al. because it is very computationally expensive [9]. We perform this experiment with German Credit only because there are no categorical features in Communities and Crime. We consider $\delta$ on only the numerical features and rounding $\delta$ to 0 or 1 for the categorical features. We present

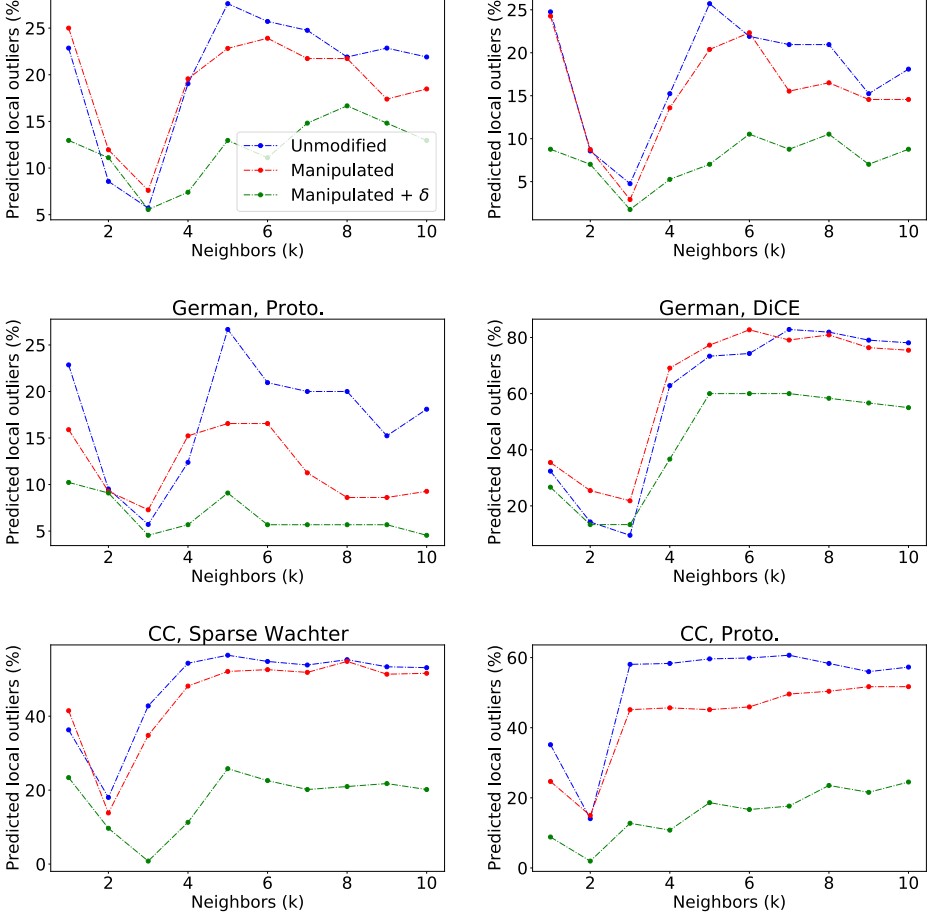

Figure 5: **Additional outlier factor results** for more data sets and counterfactual explanations indicate a similiar trend as in the main paper: the manipulated + $\delta$ counterfactuals are the most realistic (lowest % predicted local outliers).

the results in tables 5. We found the manipulation to be successful in 3 out of 4 cases, with the exception being rounded $\delta$ for counterfactuals guided by prototypes. These results demonstrate the manipulation is successful with categorical features.

Table 5: **Manipulated Models with categorical features** trained on the German Credit data set. These results show it is possible to train manipulative models successfully with categorical features.

| | Only numerical $\delta$ | | Rounding $\delta$ | |
| --- | --- | --- | --- | --- |
| | **Proto.** | **DiCE** | **Proto.** | **DiCE** |
| **German Credit** | | | | |
| Protected | 4.78 | 6.72 | 7.14 | 6.72 |
| Non-Protected | 4.19 | 5.87 | 7.17 | 5.87 |
| *Disparity* | 0.58 | 0.85 | 0.03 | 0.85 |
| Non-Protected+$\delta$ | 1.83 | 3.77 | 7.82 | 3.77 |
| *Cost Reduction* | *2.3×* | *1.5×* | *0.92×* | *1.5×* |
| Test Accuracy | 66.18%. | 68.14% | 70.0%. | 68.14% |
| $\|\delta\|_1$ | 0.79 | 0.38 | 1.92 | 0.38 |

Table 6: **Additional different initialization results for Communities & Crime** demonstrating the efficacy of different mitigation strategies. These results help demonstrate that $x$+$\mathcal{N}$ is consistently an effective mitigation strategy.

| Model | S-Wachter | | | Proto. | | |
|---|---|---|---|---|---|---|
| Initialization | Mean | Rnd. | $x$+$\mathcal{N}$ | Mean | Rnd. | $x$+$\mathcal{N}$ |
| Protected | 61.69 | 1.94 | 4.80 | 75.69 | 524.99 | 64.63 |
| Non-Protected | 44.62 | 1.71 | 2.75 | 83.98 | 503.56 | 72.08 |
| *Disparity* | *17.03* | *1.83* | *2.05* | *8.29* | *13.78* | *7.45* |
| Non-Protected+$\delta$ | 38.74 | 0.23 | 2.64 | 113.83 | 449.60 | 88.00 |
| *Cost Reduction* | *1.15×* | *7.43×* | *1.04×* | *0.74×* | *1.12×* | *0.82×* |
| Test Accuracy | 80.0%. | 79.6% | 79.9% | 81.3% | 80.7% | 80.2% |
| $||\boldsymbol{\delta}||_1$ | 0.50 | 0.51 | 0.80 | 0.68 | 0.67 | 0.68 |

Table 7: **Different initialization results for the German Credit data set** demonstrating the efficacy of various initialization strategies. These results indicate that $x$+$\mathcal{N}$ is consistently the most effective mitigation strategy.

| Model | Wachter | | | S-Wachter | | | Proto. | | | DiCE |
|---|---|---|---|---|---|---|---|---|---|---|
| Initialization | Mean | Rnd. | $x$+$\mathcal{N}$ | Mean | Rnd. | $x$+$\mathcal{N}$ | Mean | Rnd. | $x$+$\mathcal{N}$ | Rnd. |
| Protected | 1.94 | 1.18 | 1.22 | 5.58 | 0.83 | 2.18 | 3.24 | 3.81 | 3.62 | 39.53 |
| Not-Prot. | 1.29 | 1.27 | 1.42 | 2.29 | 0.95 | 3.24 | 4.64 | 7.42 | 3.47 | 36.53 |
| *Disparity* | *0.65* | *0.18* | *0.19* | *3.29* | *0.12* | *1.06* | *1.39* | *3.61* | *0.14* | *21.43* |
| Not-Prot.+$\delta$ | 0.96 | 3.79 | 1.03 | 1.30 | 1.36 | 1.26 | 3.52 | 5.74 | 2.54 | 3.00 |
| *Cost Reduction* | *1.34×* | *1.07×* | *1.38×* | *1.31×* | *0.70×* | *1.26×* | *1.32×* | *1.29×* | *1.36×* | *1.70×* |
| Accuracy | 66.5 | 67.0 | 68.5 | 66.5 | 67.7 | 67.7 | 66.3 | 65.8 | 65.8 | 66.8 |
| $||\boldsymbol{\delta}||_1$ | 0.81 | 0.80 | 0.36 | 0.81 | 0.80 | 0.54 | 0.98 | 0.43 | 0.83 | 2.9 |

# F  Compute Details

We run all experiments in this work on a machine with a single NVIDIA 2080Ti GPU.