# OpenReview forum: "Counterfactual Explanations Can Be Manipulated"
_NeurIPS.cc/2021/Conference — NeurIPS 2021 Poster_

### Official Review · Reviewer_Ky56 · 2021-07-07

**Rating:** 5
**Confidence:** 4

**Summary:**

This paper shows that algorithms used to generate counterfactual explanations are subject to manipulation. The authors provide a technique to adversarially train models that produce unstable counterfactual explanations as well as strategies to mitigate this manipulability.

**Limitations And Societal Impact:**

The authors haven't explicitly discussed potential negative impacts. Given that the paper provides techniques to learn models that look benign but have undesirable social properties, the authors should include such a discussion.

**Main Review:**

The setting being considered here is somewhat contrived, in my view. It supposes that someone building a classifier is required to ensure that the cost of recourse will be equal across two groups, and that an auditor will verify this. However, the entity building the classifier is a bad actor and wants to fool the auditor while actually producing a classifier that has a lower recourse cost for the non-protected group. I'm not convinced that "equal recourse cost" is a meaningful notion of fairness to consider, especially in the Communities and Crime dataset analyzed here. But I'm open to pushback on this.

One major issue I see is that the authors' definition of unfairness appears to require 2 things:
- Roughly equal recourse costs for 2 groups
- Substantial reduction in recourse cost for one of the groups when perturbed by $\delta$
What this doesn't cover is the change in recourse cost for the "protected" group when perturbed by $\delta$ -- if this also results in a similar reduction in recourse cost, then I don't see why this should be considered problematic. This might be possible to address by adding a term to the objective.

Beyond this, the paper is well-written and quite thorough. The experiments provided are natural, and the authors make sure to include robustness checks. They also evaluate a number of mitigation strategies.

Other small comments:
- $G(\cdot, \cdot)$ is used in Lemma 3.1 but I can't find where it's defined -- I assume it refers to the objective in Equation 2.
- It seems strange to refer to it as "Wachter's algorithm" when there are 3 authors on the paper cited.

**Time Spent Reviewing:**

2

---

> ### Author Response · Authors · 2021-08-10
> **Response to reviewer Ky56**
>
> We thank the reviewer for their useful feedback and appreciate the positive comments surrounding the quality of the writing and thoroughness of the evaluation. We respond to the reviewer’s questions below.
>
> $$\textbf{Equal Cost of Recourse}$$
>
> Equal cost of recourse is a well accepted notion of fairness in the literature [1, 2]. We apologize if the citation for the definition is not clear, and we will clarify the citation for the definition in the final paper.  Further, because of the financial incentives and predictive policing strategies tied to perceived presence of crime in communities, it is appropriate to consider recourse fairness in a setting like communities and crime [3,4]. For example, if predominantly white communities need to do less work to be predicted safe than predominantly black communities, this is unfair and could have negative impacts on the predominantly black community, such as reduced funding. Thus, this choice is well motivated. We will include this motivation when introducing the data set in the final paper.
>
> [1] Vivek Gupta, Pegah Nokhiz, Chitradeep Dutta Roy, and Suresh Venkatasubramanian. Equalizing recourse across groups. 09 2019.
>
> [2] Amir-Hossein Karimi, Gilles Barthe, Bernhard Schölkopf, and Isabel Valera. A survey of algorithmic recourse: definitions, formulations, solutions, and prospects. arXiv e-prints, art. arXiv:2010.04050, October 2020.
>
> [3] https://www.ciceroinstitute.org/performance-incentive-funding-programs
>
> [4] https://www.brennancenter.org/our-work/research-reports/predictive-policing-explained
>
> $$\textbf{Problem Setting}$$
>
> Though the reviewer indicates that the problem setting is somewhat contrived, we believe that our choice of problem setting highlights the considerable risks associated with using current counterfactual explanations and reveals fundamental flaws in how a broad set of counterfactual explanations are designed. Please note that algorithmic recourse is a nascent field with most literature focusing on coming up with new methods to generate counterfactuals. This work makes one of the first attempts at understanding how these methods can be manipulated by adversaries. As reviewer EEmF comments, “the result is very important for the reason that the limitations of counterfactual explanations must be analyzed and become well-known to prevent their widespread misuse.” In particular, our problem setting reveals how it is possible for models to hide low cost recourse, with respect to current counterfactual explanation methods, which has considerable practical implications, as also pointed out by reviewer UNQ2, "the problem raised and studied in this paper is novel and has important practical implications for the applications of counterfactual explanations."
>
> $$\textbf{Protected Group}$$
>
> The reviewer questions whether the manipulation can be considered problematic if $\delta$ applied to the protected group results in a similar cost reduction. To clarify this point, we refer to the problem setup, described in figure 2. In our problem setup, the adversary is the one who controls the model design and application of $\delta$ (note, that $\delta$ is hidden from the auditor in this figure). Because the adversary is incentivized only to provide low cost recourse for the non-protected group, they will only apply $\delta$ for members of this group. Thus, in order to generate unfair recourse, it is sufficient for the model to seem fair to the auditor (left hand side of figure 2) but have low cost recourse when the adversary applies $\delta$ to the non-protected group (right hand side of figure 2). The design of our objective and evaluation reflects this point. As the reviewer suggests, further modifications to the objective could be possible, but these are beyond the scope of the current problem setting. We will include this additional description in the problem setup to clarify this point.
>
> Small comments:
> - G is defined in equation (1).
> - We apologize for this usage and will change Wachter’s algorithm to Wachter et al.’s algorithm for fairer attribution.
>
> We additionally refer the reviewer to the comment to all reviewers, where we include a discussion about potentially negative impacts.
>
> We thank the reviewer again for their time and helpful comments. If the reviewer feels that we have adequately addressed their concerns, we would appreciate it if they consider increasing their score to “accept." We would also be very happy to provide any further details and/or clarifications.

---

### Official Review · Reviewer_qTs3 · 2021-07-12

**Rating:** 6
**Confidence:** 2

**Summary:**

The paper investigates the robustness of existing counterfactual explanation algorithms in terms of bias attributes. The method and experiments make sense.

**Limitations And Societal Impact:**

yes

**Main Review:**

The writing is clear and easy to follow and the proposed method is simple yet sound. Just curious if any idea to generalize the method to high-dimensional data like the Image domain.

-------------post rebuttal---------------

After reading other reviews, I'd like to maintain my original score.

**Time Spent Reviewing:**

2

---

> ### Author Response · Authors · 2021-08-10
> **Response to reviewer qTs3**
>
> We thank the reviewer for their useful comments. We appreciate the positive comments surrounding the exposition and soundness of the method. Thank you!
>
> We respond to the comment below. If the reviewer feels that we have adequately answered their questions on applying the method to high dimensional domains, we would appreciate it if they consider increasing their score.
>
> We did not explore a generalization to high dimensional domains like images because we believe the notion of recourse cost is difficult to define in this setting. In particular, it is difficult to understand the meaningfulness of recourse cost with respect to changes in pixel values. Further, the incentives are less clear. In a setting like lending, the financial incentives associated with loans might motivate concealing unfair advantages to certain groups. It is harder to see why an adversary would want to act in this way when it comes to images. We will add this motivation for omitting images in the discussion section of the paper.
>
> That said, our method theoretically generalizes to these domains, and it is possible to optimize the objectives discussed in section 3 with a convolutional neural network, for instance. However, the complexity of the Hessian in Lemma 3.1 scales quadratically according to the size of the input. In these settings, the Hessian may need to be approximated. This approximation could be performed by taking the diagonal of the Hessian matrix, which has been shown to work well in many settings [https://web.mit.edu/dimitrib/www/datanets.html, Chapter 5, also in B. Fernando and S. Gould. Learning end-to-end video classification with rank-pooling. ICML, 2016]. We will include this discussion concerning the complexity in the method description of the paper.

---

### Official Review · Reviewer_UNQ2 · 2021-07-22

**Rating:** 7
**Confidence:** 3

**Summary:**

This paper presents a new research problem regarding the vulnerabilities of counterfactual explanations, and introduces an adversarial model/framework to demonstrate the vulnerabilities of existing algorithms which employ hill-climbing to find counterfactual explanations (and initialize the search at the original data point). Some mitigation strategies for improving the robustness of this type of counterfactual explanation algorithms are also presented.

The work shows that the proposed adversarial model can pass the (recourse) fairness assessments which rely on counterfactual explanations produced by the counterfactual explanation methods using hill-climbing, but in fact is not a fair model as if the other features of an input instance from the non-protected group is perturbed slightly, the counterfactual explanation generated for the instance can be biased (i.e. having a much more desirable recourse). This result demonstrates the potential problem with the counterfactual explanation methods and the fairness assessment scheme based on counterfactual explanations.

**Limitations And Societal Impact:**

The authors have touched on the societal impact of the work, but since the paper proposes an adversarial model, it would be good to mention the potential negative impact of the adversarial model (e.g. someone might exploit the model/idea to design their own adversarial attacks for bad purpose).


**Main Review:**

Please find below my detailed comments on the originality, significance, quality, and clarity of the paper.

1. Originality and significance

The problem raised and studied in this paper is novel and has important practical implications for the applications of counterfactual explanations. The proposed adversarial model for demonstrating the problem is also very interesting and convincing.

2. Quality

The proposed framework/adversarial model is sound and the experiments have shown that it is possible for the manipulation to happen.

Some questions for consideration though, and some discussion on them may strengthen the paper:

(a) In the experiments feed-forward neural networks are used as adversarial models, and convincing results are obtained based on the models. I wonder how would the results would be if other types of machine learning models are used? Perhaps related to this, in theory, is it always possible or does it require similar effort for the manipulation to succeed, i.e. to obtain the "arbitrary" function f?  Some more discussions on the conditions for the success of the manipulation would be good.

(b) How sensitive the success of the manipulation is to the value of \delta? That is, how the cost reductions in Table 1 would change if we change the value of \delta (around the discovered \delta value)?

(c) Is it a practical assumption that an adversary knows which counterfactual explanation algorithm, \mathcal{A} is used by an auditor? I am not familiar with the practice, but I would guess an auditor would keep such sensitive information away from the organization being audited.

(d) Is Definition (2.1) a commonly used measure for assessing recourse fairness? If yes, a reference should be added. If not, some discussion may be needed as to why average distances are used and how the use of average distances (instead of other measures) would affect the design and success of an adversarial model.

3. Clarity

Overall the paper is well written and easy to follow. Below are some minor comments:

(a) About notation: At quite a few places, x^q and \delta are not bold-faced. Please check and correct.

(b) Line 154, page 4: the statement here about x+\delta is understandable, but stating that x+\delta should be "a" counterfactual may be confusing, as readers might get x+\delta and A(x+\delta) mixed up. I could be wrong, but I understand that as indicated in Figure 1, x+\delta does not have to be a real counterfactual as A(x+\delta) is, although the objective of the adversarial model aims to minimize the expectation shown in this line.

(c)  typos:

*Caption of Figure 1: is higher cost -> has higher cost

*Line 78, page 3: an historically -> a historically

*Line 112, Objective 2 --> Objective in Eq. (2) ?

*Line 208, page 6: State -> States

**Time Spent Reviewing:**

5

---

> ### Author Response · Authors · 2021-08-10
> **Response to Reviewer UNQ2**
>
> We thank the reviewer for their useful feedback and comments. Further, we were happy you appreciated the significance of the work, interesting nature of the method, and convincingness of the results. Thank you!
>
> We proceed by answering the reviewer’s questions and responding to the comments.
>
> $$\textbf{Conditions for Success}$$
>
> We expect our proposed framework will be effective if the model is (over) expressive and the optimization of Eqs 4 and 5 can be performed efficiently. Deep neural networks provide both, due to over-parameterization and differentiability. Although application to other model families is not within the scope of the paper, we do show how the expressiveness of the model affects the success of the attack (Figure 4C). We will include this discussion that clarifies the applicability of our approach to other model families.
>
> $$\textbf{Sensitivity of }  \delta$$
>
> We performed an experiment on Wachter et al’s algorithm for German Credit to assess the sensitivity of $\delta$. We perturb $\delta$ with noise sampled from $\mathcal{N} (0, \sigma)$ for each feature in the German credit data set and $\sigma$ is the variance ( we consider $\sigma=[1, 1e-2, 1e-4, 1e-6]$). For each value of $\sigma$, we draw $10$ different values of $\delta$ and assess the cost reduction and report the mean cost reduction. Recalling that the original cost reduction for this data set is $1.8\times$ (corresponding to $\sigma=0$), we find the mean cost reductions to be $[1.5\times, 1.13\times, 0.96\times,0.05\times]$ for $[1e-6, 1e-4, 1e-2, 1]$ respectively. These results indicate that $\delta$ is somewhat robust to small changes, while large changes, unsurprisingly, render it ineffective
>
> $$\textbf{The practicality of knowing } \mathcal{A}$$
>
> The reviewer makes a good point that in some cases, it might be reasonable to assume $\mathcal{A}$ is private. However, we believe the assumption that $\mathcal{A}$ is known is still valuable for many cases. To ensure transparency, accountability, and to more clearly define compliance with regulations, tests performed by auditing agencies are often public information. As one real world example, the EPA in the USA publishes standard tests they perform. These tests are detailed, reference the academic literature, and are freely available online (https://www.epa.gov/risk/catalogue-standard-toxicity-tests-ecological-risk-assessment). Fairness audits may likely be public information as well and thus it is reasonable to assume the used methods are generally known. We will include this in the discussion section.
>
> $$\textbf{Fairness Metric}$$
>
> This metric is a standard metric defined in Gupta et. al. [1], cited earlier in the paragraph. We apologize that this relation is not clearly indicated in the metric definition and will more prominently include the citation in this definition.
>
> Clarification points: We will incorporate these clarifications in the final version. In particular, about (b), this is the correct understanding, and we will clarify this. The typo in (c) should state the objective in equation (2).
>
> Last, we point the reviewer to the response to all the reviewers, where we add further discussion on the potential negative impacts of developing adversarial models.
>
> [1] Vivek Gupta, Pegah Nokhiz, Chitradeep Dutta Roy, and Suresh Venkatasubramanian. Equalizing recourse across groups. 09 2019.

---

> > ### Comment · Reviewer_UNQ2 · 2021-08-31
> > **Thanks for the response**
> >
> > Thank you very much for the detailed responses/discussions. They are very helpful for me to understand the relevant details. I will keep my positive score on the paper.

---

> > > ### Author Response · Authors · 2021-09-07
> > > **Thank you!**
> > >
> > > We are very glad to know that the reviewer found our rebuttal helpful. Thank you so much for keeping your positive score.

---

### Official Review · Reviewer_EEmF · 2021-08-02

**Rating:** 7
**Confidence:** 4

**Summary:**

Context: Counterfactual explanations are a way to locally interpret model decisions for a given input by finding a "nearby" input with a different model decision. As in this paper, we can interpret some distance measure between the original input and its "counterfactual explanation" point as a kind of cost of achieving a different decision, usually for/to an individual who gets a negative decision to understand what they would need to change to receive a positive decision. In a fairness context we can consider differences in such costs between different groups as evidence of unfairness, for example if there are higher costs (on average, say) for individuals from a protected non-discrimination group.

The current paper shows that for a certain class of algorithms producing counterfactual explanations, it is possible for a small, adversarial perturbation of the input to produce a dramatically different explanation. The paper points out that this non-robustness of the algorithms in consideration is important from a fairness or regulatory standpoint by simulating a scenario where a model builder can jointly train a classifier and adversarial perturbation so that an auditor relying on counterfactual explanations (using any of the algorithms in consideration here) can be deceived. The paper shows how to train such models that can hide unfairness by providing a low-cost explanation for a nearby point relative to the higher cost explanation of the original input, and demonstrates the concept with experiments using two datasets and four different counterfactual explanation algorithms in the literature. Experiments also show how to achieve robustness by changing the initialization point for the counterfactual explanation algorithm, by reducing the dimension of the input space, or by otherwise constraining the complexity of the classification model.

**Limitations And Societal Impact:**

No. In addition to my first point under the "Suggestions" above, this paper also uses a crime prediction dataset for one of its experiments without acknowledging the potential dangers and shortcomings of such a task that have been extensively critiqued in other algorithmic fairness papers.

**Main Review:**

Originality: This paper adds to a literature of others discussing the shortcomings of so-called counterfactual explanations, and to the list of papers showing that adversarial training can reveal shortcomings of a given class of algorithms. But I believe the combination/intersection is original, i.e. to my knowledge it is the first to directly apply adversarial training in such a setup to produce explanations with lower cost and thereby hide unfairness.

Quality: The results are sound and well-supported by the experiments, the work is mature and relatively complete, and the paper is honest about its assumptions and limitations.

Clarity: The paper is clearly written and informative.

Significance: The result is very important for the reason that the limitations of counterfactual explanations must be analyzed and become well-known to prevent their widespread misuse.

Suggestions: In my opinion there are two things that could improve the paper. The first, and most important, would be to shift the fairness part of the focus away from "how to train a model that can hide its unfairness" and toward "how to determine if a model is hiding unfairness." The best version of such a shift would require significant revision, possibly including changes of the auditing setup to require different access. The problem is that the paper currently has in-depth instruction on how model owners with bad motives could do something destructive and unethical, with a brief follow-up of some methods that model owners or auditors with good motives could try to avoid this problem. My second suggestion is a minor one and concerns the term "counterfactual explanation." It is an unfortunate fact that a sizeable literature has already adopted this term despite the fact that it misuses the word "counterfactual." There is no explicit causal model or even philosophical assumption from which these explanations justify their use of the word, it is just comparing one input point to another input point. Several of the papers on the limitations of this approach point out this fact, including ones cited in the current paper. I think it might be helpful if the current paper (and any others that continue misusing the term "counterfactual") have some disclaimer early on (e.g. in the introduction) regarding the absence of an actual causal model justification for using the word. This could help some readers avoid confusion.

**Time Spent Reviewing:**

5

---

> ### Author Response · Authors · 2021-08-10
> **Author response to Reviewer EEmF**
>
> We thank the reviewer for their thoughtful and helpful comments. We were delighted you appreciate the importance of the work, quality in presentation, and overall clarity. Thank you! Below we address the specific questions raised by the review.
>
> $$\textbf{Focus of the Paper}$$
>
> We thank the reviewer for their comment. While we acknowledge that determining if the model is hiding unfairness is a critical problem, we believe that it is important to first thoroughly understand how an adversary can manipulate these explanations. Once we have a clear understanding of how this can be done, then we can focus on the next step of devising strategies to determine if a model is hiding unfairness. Please note that algorithmic recourse is a nascent field with most literature focusing on coming up with new methods to generate counterfactuals. This work makes one of the first attempts at understanding how these methods can be manipulated by adversaries. We believe that this work takes the initial step towards providing insights about how counterfactual explanations can be potentially misused, and thereby motivate further work on testing if counterfactual explanations are hiding unfairness, and how to design counterfactual explanations robust to such manipulations.
>
> $$\textbf{Counterfactual Term}$$
>
> We appreciate the suggestion and will add discussion to clarify the usage conventions of the term “counterfactual explanation” in the paper. In particular, we will discuss how prior works use the terms counterfactual explanations [1], contrastive explanations [2], and recourse [3] interchangeably to describe ways to provide recourse to individuals with unfavorable algorithmic decisions and that many of these works do not assume access to a causal model, as is the same in our work.
>
> We additionally refer the reviewer to the comment for all the reviewers where we include a disclaimer surrounding violent crime prediction in communities.
>
> [1]  Sandra Wachter, Brent Mittelstadt, and Chris Russell. Counterfactual explanations without opening the black box: Automated decisions and the gdpr. Harvard journal of law & technology, 31:841–887, 04 2018. doi: 10.2139/ssrn.3063289.
>
> [2] Amir-Hossein Karimi, Gilles Barthe, Bernhard Schölkopf, and Isabel Valera. A survey of algorithmic recourse: definitions, formulations, solutions, and prospects. arXiv e-prints, art. arXiv:2010.04050, October 2020.
>
> [3]  Berk Ustun, Alexander Spangher, and Yang Liu. Actionable recourse in linear classification. In Proceedings of the Conference on Fairness, Accountability, and Transparency, FAT* ’19, pages 10–19. ACM, 2019. ISBN 978-1-4503-6125-5. doi: 10.1145/3287560.3287566. URL http://doi. acm.org/10.1145/3287560.3287566.

---

### Author Response · Authors · 2021-08-10
**General Response**

We thank all the reviewers for their thoughtful comments and valuable feedback. We appreciate the comments among reviewers that the paper is well written, the experiments are thorough, the ideas are original, the results are significant, and the paper has important practical implications for counterfactual explanations in the future.

We respond to each of the reviewers' questions individually. However, one point shared across reviewers was that the paper lacks discussion surrounding the potential negative implications of developing adversarial models. Additionally, one reviewer raised the point that the fairness community has extensively critiqued crime prediction tasks. Though we include a brief discussion of impacts in the checklist section, we appreciate the comments and will include such a discussion in the final paper, along the following lines:

$$\textbf{Impacts of developing adversarial models}$$

Our goal in designing adversarial models is to demonstrate how counterfactual explanations can be misused, and in this way, prevent such occurrences in the real world, either by informing practitioners of the risks associated with their use or motivating the development of more robust counterfactual explanations. However, there are some risks that the proposed techniques could be applied to generate manipulative models that are used for harmful purposes. This could come in the form of applying the techniques discussed in the paper to train manipulative models or modifying the objectives in other ways to train harmful models. However, exposing such manipulations is one of the key ways to make designers of recourse systems aware of risks so that they can ensure that they place appropriate checks in place and/or design robust counterfactual generation algorithms.

$$\textbf{Critiques of crime prediction tasks}$$

In the paper, we also include the Communities and Crime data set.  The goal of this data set is to predict whether violent crime occurs in communities. Using machine learning in the contexts of criminal justice and crime prediction has been extensively critiqued by the fairness community [1, 2, 3]. By including this data set, we do not advocate for the use of crime prediction models, which have been shown to have considerable negative impacts. Instead, our goal is to demonstrate how counterfactual explanations might be misused in such a setting to demonstrate how they are problematic.

We again thank the reviewers for their time and valuable comments.

[1] Dressel, Julia and Farid, Hany. The accuracy, fairness, and limits of predicting recidivism. American Association for the Advancement of Science. 2018.

[2] Julia Angwin, Jeff Larson, Surya Mattu, and Lauren Kirchner. Machine bias. In ProPublica, 2016.

[3] Rudin, C., Wang, C., & Coker, B. (2020). The Age of Secrecy and Unfairness in Recidivism Prediction. Harvard Data Science Review, 2(1). https://doi.org/10.1162/99608f92.6ed64b30

---

### Decision · Program_Chairs · 2021-09-27

**Decision:**

Accept (Poster)

**Comment:**

This paper contributes to the growing literature documenting the instability and manipulability of different model explanation techniques.  In this work the authors specifically study counterfactual explanations, which commonly describe how an individual's inputs could change to receive a positive classification.  Overall the reviewers agree that the specific problem formulation tackled in this work is novel and that the work improves our understanding of how models can be adversarially manipulated to produce biased recourse costs by hiding low cost recourse paths.  Reviewers also agreed that the paper is well-developed, overall clearly written, and comprehensive.

The authors have already proposed additional text discussing the implications of developing adversarial models and issues with the common practice of relying on crime-related data in experiments.  Beyond this, the set of revisions in preparation for the camera-ready is fairly small.  For readers who are unfamiliar with the prior literature on "counterfactual" explanations, I agree with Reviewer EEmF that it would be work clarifying that the term "counterfactual" doesn't have the same meaning here as it does in the context of causal inference. Other reviewers provided lists of requested clarifications/modifications that are all actionable and would improve the clarity and accuracy of the manuscript.

Congratulations on an excellent submission!